# VidProM : A Million-scale Real Prompt-Gallery Dataset for Text-to-Video Diffusion Models

**Wenhao Wang**
University of Technology Sydney
wangwenhao0716@gmail.com

**Yi Yang***
Zhejiang University
yangyics@zju.edu.cn

**1.67 Million Unique Text-to-Video Prompts from Real Users**

*Create a vivid 3D village scene in the midst of a journey, where quaint cottages dot the landscape, surrounded by lush greenery*

*Take the viewers on a captivating cinematic journey in 8K ultra HD, following a group of chicks in Disney style as they explore a magical forest. They play until the sun sets, and the lush, enchanting environment enchants the audience. Convey the idea that time ceases to exist as they immerse themselves in their playful adventures.*

*a Chinese dragon flying in the city sky, large scene*

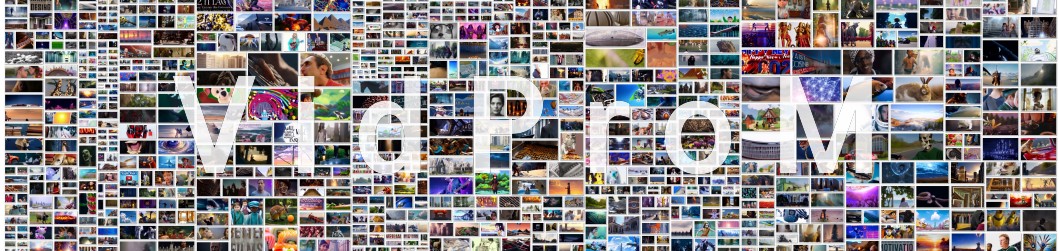

**6.69 Million Generated Videos by Diffusion Models**

Figure 1: VidProM is the first dataset featuring 1.67 million unique text-to-video prompts and 6.69 million videos generated from 4 different state-of-the-art diffusion models. It inspires many exciting new research areas, such as Text-to-Video Prompt Engineering, Efficient Video Generation, Fake Video Detection, and Video Copy Detection for Diffusion Models.

## Abstract

The arrival of Sora marks a new era for text-to-video diffusion models, bringing significant advancements in video generation and potential applications. However, Sora, along with other text-to-video diffusion models, is highly reliant on *prompts*, and there is no publicly available dataset that features a study of text-to-video prompts. In this paper, we introduce **VidProM**, the first large-scale dataset comprising 1.67 **M**illion unique text-to-**Vid**eo **Pro**mpts from real users. Additionally, this dataset includes 6.69 million videos generated by four state-of-the-art diffusion models, alongside some related data. We initially discuss the curation of this large-scale dataset, a process that is both time-consuming and costly. Subsequently, we underscore the need for a new prompt dataset specifically designed for text-to-video generation by illustrating how VidProM differs from DiffusionDB, a large-scale prompt-gallery dataset for image generation. Our extensive and diverse dataset also opens up many exciting new research areas. For instance, we suggest

---

*Corresponding Author.

38th Conference on Neural Information Processing Systems (NeurIPS 2024) Track on Datasets and Benchmarks.

exploring text-to-video prompt engineering, efficient video generation, and video copy detection for diffusion models to develop better, more efficient, and safer models. The project (including the collected dataset VidProM and related code) is publicly available at https://vidprom.github.io under the CC-BY-NC 4.0 License.

# 1  Introduction

The Sora [1] initializes a new era for text-to-video diffusion models, revolutionizing video generation with significant advancements. This breakthrough provides new possibilities for storytelling, immersive experiences, and content creation, as Sora [1] transforms textual descriptions into high-quality videos with ease. However, Sora [1] and other text-to-video diffusion models [2, 3, 4, 5] heavily relies on the prompts used. Despite their importance, there is no publicly available dataset focusing on text-to-video prompts, which may hinder the development of these models and related researches.

In this paper, we present the first systematic research on the text-to-video prompts. Specifically, our efforts primarily focus on building the first text-to-video prompt-gallery dataset VidProM, analyzing the necessity of collecting a new prompt dataset specialized for text-to-video diffusion models, and introducing new research directions based on our VidProM. The demonstration of VidProM is shown in Fig. 1.

• **The first text-to-video prompt-gallery dataset.** Our large-scale VidProM includes 1.67 million unique text-to-video prompts from real users and 6.69 million generated videos by 4 state-of-the-art diffusion models. The prompts are from official Pika Discord channels, and the videos are generated by Pika [2], Text2Video-Zero [3], VideoCraft2 [4], and ModelScope [5]. We distribute the generation process across 10 servers, each equipped with 8 Nvidia V100 GPUs. Each prompt is embedded using the powerful text-embedding-3-large model from OpenAI and assigned six not-safe-for-work (NSFW) probabilities, consisting of toxicity, obscenity, identity attack, insult, threat, and sexual explicitness. We also add a Universally Unique Identifier (UUID) and a time stamp to each data point in our VidProM. In addition to the main dataset, we introduce a subset named VidProS, which consists of *semantically unique* prompts. That means, in this subset, the cosine similarity between any two prompts is less than 0.8, ensuring a high level of semantic diversity.

• **The necessity of collecting a new prompt dataset specialized for text-to-video diffusion models.** We notice that there exists a text-to-image prompt-gallery dataset, DiffusionDB [6]. By analyzing the basic information and the prompts, we conclude that the differences between our VidProM and DiffusionDB [6] mainly lies in: (1) Semantics: The semantics of our prompts are significantly different from those in DiffusionDB, with our text-to-video prompts generally being more dynamic, more complex, and longer. (2) Modality: DiffusionDB [6] focuses on *images*, while our VidProM is specialized in *videos*. (3) Techniques: We utilize some recent advanced techniques, such as latest text embedding model (OpenAI-text-embedding-3-large), to build our VidProM.

• **Inspiring new research directions.** The introduction of our new text-to-video prompt-gallery dataset, VidProM, opens up many exciting research directions. With the help of our VidProM, researchers can develop better, more efficient, and safer text-to-video diffusion models: (1) For better models, researchers can utilize our VidProM as a comprehensive set of prompts to evaluate their trained models, distill new models using our prompt-(generated)-video pairs, and engage in prompt engineering. (2) For more efficient models, researchers can search for related prompts in our VidProM and reconstruct new videos from similar existing videos, thereby avoiding the need to generate videos from scratch. (3) For safer models, researchers can develop specialized models to distinguish generated videos from real videos to combat misinformation, and train video copy detection models to identify potential copyright issues.

To sum up, this paper makes the following contributions: **(1)** We contribute the first text-to-video prompt-gallery dataset, VidProM, which includes 1.67 million unique prompts from real users and 6.69 million generated videos by 4 state-of-the-art diffusion models. **(2)** We highlight the necessity of collecting a new prompt dataset specialized for text-to-video diffusion models by comparing our VidProM with DiffusionDB. **(3)** We reveal several exciting research directions inspired by VidProM and position it as a rich database for future studies.

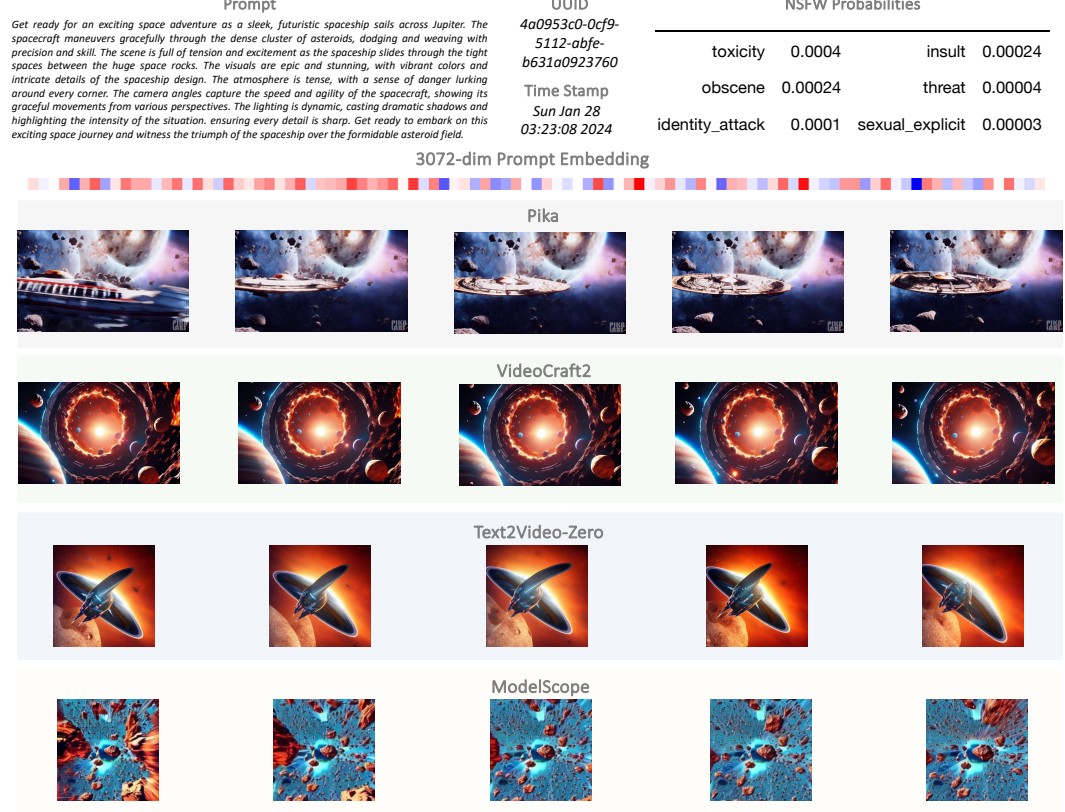

Figure 2: A data point in the proposed VidProM. Please click the corresponding links to view the complete videos: Pika, VideoCraft2, Text2Video-Zero, and ModelScope. To better understand our dataset, yon can also download 10, 000 randomly-selected data points from here.

## 2 Related Works

### 2.1 Text-to-Video Diffusion Models

Text-to-video diffusion models [3, 7, 4, 8, 5, 1, 9, 10, 2, 11, 12] have become a powerful tool for producing high-quality video content from textual prompts. *Pika* [2] is a commercial text-to-video model by Pika Labs, which advances the field of video generation. *Text2Video-Zero* [3] enables zero-shot video generation using textual prompts. *VideoCrafter2* [4] generates videos with high visual quality and precise text-video alignment without requiring high-quality videos. *ModelScope* [5] evolves from a text-to-image model by adding spatio-temporal blocks for consistent frame generation and smooth movement transitions. This paper uses these four publicly accessible sources (access to generated videos or pre-trained weights) for constructing our VidProM. We hope that the collection of diffusion-generated videos will be useful for further research in text-to-video generation community.

### 2.2 Existing Datasets

**Text-Video Datasets.** While several published text-video datasets exist [13, 14, 15, 16, 17, 18, 19, 20], they primarily consist of caption-(real)-video pairs rather than prompt-(generated)-video pairs. For example, *WebVid-10M* is a large-scale text-video dataset with 10 million video-text pairs collected from stock footage websites [16]. *HDVILA-100M* [17] is a comprehensive video-language dataset designed for multimodal representation learning, offering high-resolution and diverse content. *Panda-70M* [18] is a curated subset of HDVILA-100M [17], featuring semantically consistent and high-resolution videos. In contrast, our VidProM contains prompts authored by real users to generate videos of interest, and the videos are produced by text-to-video diffusion models.

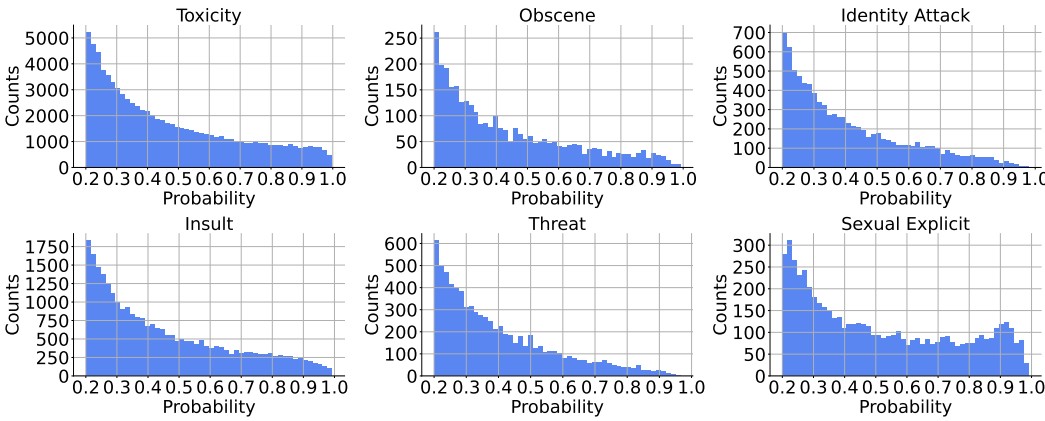

Figure 3: The number of prompts with an NSFW probability greater than $0.2$ constitutes only a very small fraction of our total of $1,672,243$ unique prompts.

**Prompt Datasets.** Existing datasets underscore the significance of compiling a set of prompts. In the *text-to-text* domain, studies [21] demonstrate that gathering and analyzing prompts can aid in developing language models that respond more effectively to prompts. *PromptSource* [22] recognizes the growing popularity of using prompts to train and query language models, and thus create a system for generating, sharing, and utilizing natural language prompts. In the *text-to-image* domain, *DiffusionDB* [6], which is nominated for the best paper of ACL 2023, collects a large-scale prompt-image dataset, revealing its potential to open up new avenues for research. Given the importance of prompt datasets and the new era of text-to-video generation brought by *Sora* [1], this paper presents the first prompt dataset specifically collected for *text-to-video* generation.

## 3 Curating VidProM

In Fig. 2, we illustrate a single data point in the proposed VidProM. This data point includes a prompt, a UUID, a timestamp, six NSFW probabilities, a 3072-dimensional prompt embedding, and four generated videos. We show the steps of curating our **VidProM** in this section.

**Collecting Source HTML Files.** We gather chat messages from the official Pika Discord channels between July 2023 and February 2024 using DiscordChatExporter [23] and store them as HTML files. Our focus is on 10 channels where users input prompts and request a bot to execute the Pika text-to-video diffusion model for video generation. The user inputs and outputs are made available by Pika Lab under the Creative Commons Noncommercial 4.0 Attribution International License (CC BY-NC 4.0), as detailed in Section 4.5.a of their official terms of service. Consequently, the text-to-video prompts and Pika videos in our dataset are open-sourced under the same license.

**Extracting and Embedding Prompts.** The HTML files are then processed using regular expressions to extract prompts and time stamps. We subsequently filter out prompts used for image-to-video generation (because the images are not publicly available) and prompts without associated videos (these prompts may have been banned by Pika or hidden by the users). Finally, we remove duplicate prompts and assign a UUID to each prompt, resulting in a total of $1,672,243$ unique prompts. Because the text-to-video prompts are significantly complex and long, we use OpenAI's text-embedding-3-large API, which supports up to $8192$ tokens, to embed all of our prompts. We retain the original 3072-dimensional output, allowing any customized dimensionality reduction.

**Assigning NSFW Probabilities.** We select the public Discord channels of Pika Labs, which prohibit NSFW content, as the source for our text-to-video prompts. Consequently, if a user submits a harmful prompt, the channel will automatically reject it. However, we find VidProM still includes NSFW prompts that were not filtered by Pika. We employ a state-of-the-art NSFW model, Detoxify [24], to assign probabilities in six aspects of NSFW content, including toxicity, obscenity, identity attack, insult, threat, and sexual explicitness, to each prompt. In Fig. 3, we visualize the number of prompts with a NSFW probability greater than $0.2$. We conclude that only a very small fraction (less than

Table 1: The comparison of basic information of VidProM and DiffusionDB [6]. To ensure a fair comparison of semantically unique prompts, we use the text-embedding-3-large API to re-embed prompts in DiffusionDB [6].

| Aspects | Details | DiffusionDB [6] | VidProM |
|---|---|---|---|
| Prompts | No. of unique prompts | $1,819,808$ | $1,672,243$ |
| | No. of semantically unique prompts | $739,010$ | $1,038,805$ |
| | Embedding of prompts | OpenAI-CLIP | OpenAI-text-embedding-3-large |
| | Maximum length of prompts | 77 tokens | 8192 tokens |
| | Time span | Aug 2022 | Jul 2023 $\sim$ Feb 2024 |
| Images or Videos | No. of images/videos | $\sim 14$ million images | $\sim 6.69$ million videos |
| | No. of sources | 1 | 4 |
| | Average repetition rate per source | $\sim 8.2$ | 1 |
| | Collection method | Web scraping | Web scraping + Local generation |
| | GPU consumption | - | $\sim 50,631$ V100 GPU hours |
| | Total seconds | - | $\sim 14,381,289.8$ seconds |

$0.5\%$) of prompts have a probability greater than $0.2$. We provide six separate NSFW probabilities, enabling researchers to set a suitable threshold for filtering out potentially unsafe data for their tasks.

**Scraping and Generating Videos.** We enhance our VidProM diversity by not only scraping Pika videos from extracted links but also utilizing three state-of-the-art open-source text-to-video diffusion models for video generation. This process demands significant computational resources: we distribute the text-to-video generation across 10 servers, each equipped with 8 Nvidia V100 GPUs. It costs us approximately 50,631 GPU hours and results in 6.69 million videos ($4 \times 1,672,243$), totaling 14,381,289.8 seconds in duration. The breakdown of video lengths is as follows: 3.0 seconds for Pika, 1.6 seconds for VideoCraft2, 2.0 seconds for Text2Video-Zero, and 2.0 seconds for ModelScope.

**Selecting Semantically Unique Prompts.** Beyond general uniqueness, we introduce a new concept: semantically unique prompts. We define a dataset as containing only semantically unique prompts if, for any two arbitrary prompts, their cosine similarity calculated using text-embedding-3-large embeddings is less than $0.8$. After semantic de-duplication (see a detailed description of this process in the Appendix (Section A)), our VidProM still contains $1,038,805$ semantically unique prompts, and we denote it as VidProS. More semantically unique prompts imply covering a broader range of topics, increasing the diversity and richness of the content.

## 4   The Necessity of Introducing VidProM

We notice that there exists a text-to-image prompt-gallery dataset, DiffusionDB [6]. This section highlights how our VidProM is different from this dataset from two aspects, *i.e.* basic information and semantics of prompts.

### 4.1   Basic Information

In Table 1, we provide a comparison of the basic information between our VidProM and DiffusionDB [6]. We have the following observations:

**Prompt. (1)** Although the total number of unique prompts in our VidProM and DiffusionDB are similar, VidProM contains **significantly more** ($+40.6\%$) semantically unique prompts. This shows VidProM is a more diverse and representative dataset. **(2)** Unlike DiffusionDB, which uses the OpenAI-CLIP embedding method, our approach leverages the **latest** OpenAI text-embedding model, namely text-embedding-3-large. One advantage of this approach is its ability to accept much longer prompts compared to CLIP, supporting up to 8192 tokens versus CLIP's 77 tokens. As illustrated by the comparison of the number of words per prompt between VidProM and DiffusionDB in Fig. 4, the prompts used for generating videos are much more longer. Therefore, the capability of text-embedding-3-large is particularly suitable for them. Another advantage is text-embedding-3-large has stronger performance than CLIP on several standard benchmarks, potentially benefiting users of our VidProM. **(3)** The time span for collecting prompts in VidProM is **much longer** than in DiffusionDB. We collect prompts written by real users over a period of 8 months, while DiffusionDB's collection spans only 1 month. A longer collection period implies a broader range of topics and themes covered, as demonstrated by the comparison of the number of semantically unique prompts.

**Prompts from DiffusionDB**

*halloween scary castle, pumpkin, bar, spiders, grave, web*

*earth from outer space, cosmic dust, nebula, star, other planets, highly detailed, high render,*

*photo of chris cornell holding a kitty*

*abstract watercolor painting of spanish street, white buildings, summer, magical and traditional, cinematic light, french cafe, sharp shadows, daylight, national romanticism by anders zorn, by greg rutkowski, by greg manchess*

*a sandwich car painting a balloon made of coke*

*It had been flying for a long time and was feeling extremely parched. Finally, after a while, it spotted a small village with a few houses and trees.*

*zoom in Steampunk Android  attached to wires cables  Beksinski  Alfred Stieglitz rolling shutter vhs glitch grainy black and white 1900s*

*This is a 1-minute realistic short video depicting a young person developing good habits by using a height-adjustable automated lift desk for work in a bright and minimalist bedroom. The true-to-life footage vividly shows him working on the computer, writing at the desk, as well as clips of him standing up for exercise. The concise narration explains the practical benefits of cultivating healthy work and lifestyle habits with this type of smart furniture.*

**Prompts from our VidProM**

(a)

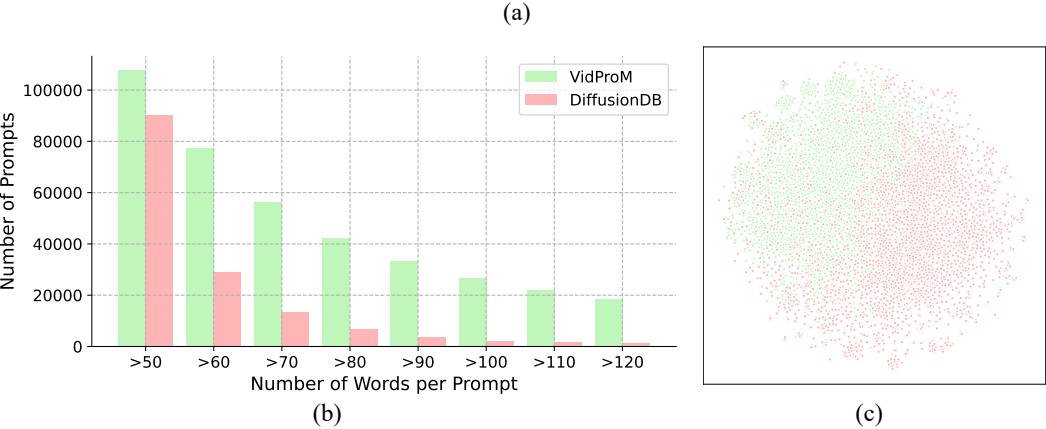

(b)        (c)

Figure 4: The differences between the prompts in DiffusionDB and our VidProM are illustrated by: (a) a few example prompts for illustration; (b) the number of long prompts; and (c) the t-SNE [25] visualization of 10,000 prompts randomly selected from DiffusionDB and VidProM, respectively.

> **Takeaway:** Our VidProM dataset contains a larger number of semantically unique prompts, which are embedded by a more advanced model and collected over a longer period.

**Images or videos.** DiffusionDB focuses on images, while our VidProM is specialized in **videos**. Therefore, given that generating videos is much more expensive than images, it is reasonable that the number of videos in VidProM is smaller than the number of images in DiffusionDB. We also make several efforts to mitigate this disadvantage: **(1)** The number of source diffusion models for our VidProM is **much larger** than those of DiffusionDB. Our videos are generated by 4 state-of-the-art text-to-video diffusion models, while DiffusionDB contains only images generated by Stable Diffusion. As a result, the average repetition rate per source is only 1 for our VidProM compared to about 8.2 for DiffusionDB. **(2)** We devote **significantly more resources** to VidProM. Unlike DiffusionDB, which only collects images through web scraping, we also deploy three open-source text-to-video models on our local servers, dedicating over 50,000 V100 GPU hours to video

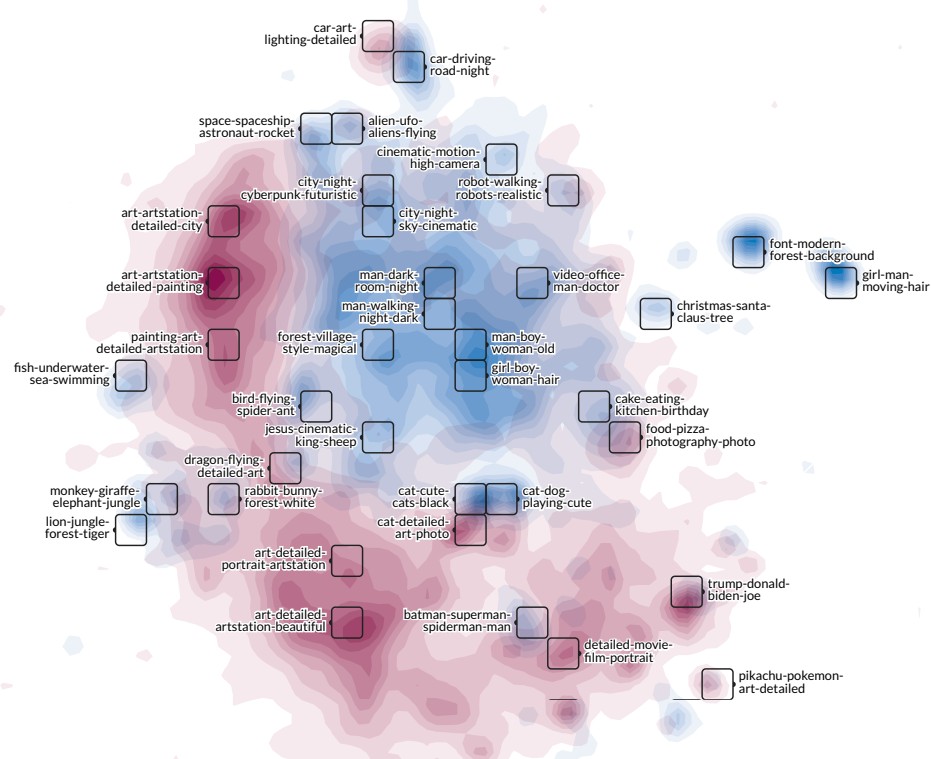

Figure 5: The WizMap [26] of our VidProM and DiffusionDB [6]. Click here for an interactive one.

generation. These efforts result in more than 14 million seconds of video. In the future, researchers can also generate more videos using the prompts with more advanced text-to-video diffusion models.

> **Takeaway:** Our VidProM dataset contains a considerable amount of videos generated by various state-of-the-art text-to-video diffusion models, utilizing a substantial amount of resources.

## 4.2 Semantics of prompts

In this section, we analyze how the semantics of prompts in our VidProM dataset are different from DiffusionDB [6].

**Firstly,** as shown in Fig. 4 (a), the semantics of the prompts differ in three aspects: (1) **Time dimension**: text-to-video prompts usually need to include a description of the time dimension, such as 'changes in actions' and 'transitions in scenes'; while text-to-image prompts typically describe a scene or object. (2) **Dynamic description**: text-to-video prompts often need to describe the dynamic behavior of objects, such as 'flying', 'working', and 'writing'; while text-to-image prompts focus more on describing the static appearance of objects. (3) **Duration**: text-to-video prompts may need to specify the duration of the video or an action, such as 'a long time' and '1-minute', while text-to-image prompts do not need to consider the time factor.

**Secondly,** as shown in Fig. 4 (a) and (b), text-to-video prompts are generally **more complex and longer** than text-to-image prompts, due to the need to describe additional dimensions and dynamic changes. This phenomenon is also observed in the prompts used by **Sora**. For instance, the prompt for the 'Tokyo Girl' video contains 64 words, while the longest prompt on the OpenAI official website comprises 95 words. Our VidProM dataset prominently features this characteristic: (1) the number of prompts with more than 70 words is nearly 60,000 for our VidProM, compared to only about 15,000 for DiffusionDB; and (2) our VidProM still has over 25,000 prompts with more than 100 words, whereas this number is close to 0 for DiffusionDB.

**Finally,** as shown in Fig. 4 (c) and Fig. 5, the prompts from our VidProM dataset and DiffusionDB exhibit **different distributions**. We use the text-embedding-3-large model to re-extract the features of the prompts in DiffusionDB. For visualizing with t-SNE [25], we randomly select $10,000$ prompt features from both our VidProM dataset and DiffusionDB, respectively. We find that these prompts have significantly different distributions and are nearly linearly separable. Beyond the traditional t-SNE [25], we use a recent method, WizMap [26], to analyze the topics preferred by people using all prompts from our VidProM and DiffusionDB. WizMap [26] is a visualization tool to navigate and interpret large-scale embedding spaces with ease. From the visualization results shown in Fig. 5, we conclude that, **despite some overlaps, there are distinct differences in their interests.** For example, both groups are interested in topics like cars, food, and cute animals. However, text-to-image users tend to focus on generating art paintings, whereas text-to-video users show little interest in this area. Instead, text-to-video users are more inclined to create general human activities, such as walking, which are rarely produced by text-to-image users.

> **Takeaway:** The significant difference in semantics between text-to-image prompts and our text-to-video prompts indicates the need to collect a new dataset of prompts specifically for video generation. By analyzing prompt usage distribution, future researchers can design generative models to better cater to popular topics in prompts.

## 5 Inspiring New Research

The proposed million-scale VidProM dataset inspires new directions for researchers to develop better, more efficient, and safer text-to-video diffusion models.

**Video Generative Model Evaluation** aims to assess the performance and quality of text-to-video generative models. Current evaluation efforts, such as [27, 28, 29, 30], are conducted using carefully designed and small-scale prompts. *Our VidProM dataset* brings imagination to this field: (1) Instead of using carefully designed prompts, researchers could consider whether their models can generalize to prompts from real users, which would make their evaluation more practical. (2) Performing evaluations on large-scale datasets will make their arguments more convincing.

**Text-to-Video Diffusion Model Development** aims to create diffusion models capable of converting textual descriptions into dynamic and realistic videos. The current methods [3, 7, 4, 8, 5, 1, 9, 10, 2, 11, 12] are trained on caption-(real)-video pairs. Two natural questions arise: (1) Will the domain gap between captions and prompts from real users (see the evidence of existing domain gap in the Appendix (Section B)) hinder these models' performance? For instance, $60\%$ of the training data for Open-Sora-Plan v1.0.0 [31] consist of landscape videos, leading to the trained model's suboptimal performance in generating scenes involving humans and cute animals. (2) Can researchers train or distill new text-to-video diffusion models on prompt-(generated)-video pairs? Although some studies indicate that this approach may lead to irreversible defects [32], as we will likely run out of data on websites and some methods are proposed to prevent such collapse [33], exploring synthetic data remains a valuable endeavor. The studies of training text-to-video diffusion models *with our VidProM* may provide answers to these two questions.

**Text-to-Video Prompt Engineering** is to optimize the interaction between humans and text-to-video models, ensuring that the models understand the task at hand and generate relevant, accurate, and coherent videos. The prompt engineering field has gained attention in large language models [34, 35], text-to-image diffusion models [36, 37], and visual in-context learning [38, 39]. However, as far as we know, there is no related research in the text-to-video community. *Our VidProM* provides an abundant resource for text-to-video prompt engineering. In the Section 6, we train a large language model on our VidProM for automatic text-to-video prompt completion for instance.

**Efficient Video Generation.** The current text-to-video diffusion models are very time-consuming. For example, on a single V100 GPU, ModelScope [5] requires 43 seconds, while VideoCrafter2 [4] needs 51 seconds to generate a video, respectively. *Our large-scale VidProM* provides a unique opportunity for efficient video generation. Given an input prompt, a straightforward approach is to search for the most closely related prompts in our VidProM and reconstruct a video from the corresponding existing videos, instead of generating a new video from noise.

**Fake Video Detection** aims to distinguish between real videos and those generated by diffusion models. While there are some works [40, 41, 42, 43, 44, 45, 46, 47, 48, 49] focusing on fake image detection, fake video detection presents unique challenges: (1) The generalization problem: Existing fake image detectors may not generalize well to video frames (see experiments in the Appendix (Section C)). For instance, a model trained on images generated by Stable Diffusion [50] may fail to identify frames from videos generated by Pika [2] or Sora [1] as fake. (2) The efficiency problem: Currently, there is no detector that can take an entire video as input. As a result, to achieve higher accuracy, we may use fake image detectors to examine all or representative frames, which can be time-consuming. *With our VidProM*, researchers can (1) train specialized Fake Video Detection models on millions of generated videos, and (2) use millions of prompts to generate more videos from more diffusion models to further improve the detection performance.

**Video Copy Detection for Diffusion Models** aims to answer whether videos generated by diffusion models replicate the contents of existing ones (see a detailed discussion in the Appendix (Section D)). Videos generated with replicated content may infringe on the copyrights of the original videos. However, current video copy detection researches [51, 52, 53, 54] focus on detecting hand-crafted copies generated by transformations like horizontal flips and random crops, and overlook the challenge from diffusion models. Researchers can utilize *our dataset VidProM* as input queries and employ existing video copy detection models to pre-filter potential replications. Subsequently, these filtered videos can be manually annotated to determine whether they are indeed replications and regarded as training sources.

# 6    Automatic Text-to-video Prompt Completion

This section describes the training and output of our text-to-video prompt completion model.

**Training details.**    Similar to text2image-prompt-generator [55], we treat the automatic text-to-video prompt completion as a general text generation task.    The training code is available at: `https://github.com/huggingface/transformers/blob/main/examples/pytorch/language-modeling/run_clm.py`. We fine-tune the recent state-of-the-art large language model, Mistral-7B-v0.1 [56], on our dataset VidProM. The fine-tuning process is completed using DeepSpeed [57] with 8 A100 GPUs in 2 hours.

**Demonstration.** We release our trained model on: `https://huggingface.co/WenhaoWang/AutoT2VPrompt`. It can be freely downloaded and tested under CC BY-NC 4.0 License, and we also demonstrate some of its generation across various topics below.

---

**Input:** A cat sitting
**Generated prompts:**
1. A cat sitting on a desk, typing on its laptop. The room has soft, natural light from the window, casting a wm glow on the cat.
2. A cat sitting in a tree, blinking its eyes. The cat is talking to the camera.. a woman is eating a hotdog in the street, sunset .
3. A cat sitting at an office desk, facing the camera. The cat has a human body shape. The cat is black in color, and the desk has some office supplies. The scene is shot at an angle.

---

**Input:** An underwater world
**Generated prompts:**
1. An underwater world. The ocean is filled with bioluminess as the water reflects a soft glow from a bioluminescent phosphorescent light source. The camera slowly moves away and zooms in.
2. An underwater world, the ocean is full of discded items, the water flows, and the light penetrating through the water.
3. An underwater world, 25 ye boy, with aqua-green eyes, dk sandy blond hair, from the back, and on his back a fish, 23 ye old, weing glasses,ctoon chacte.

---

> **Input:** A landscape
> **Generated prompts:**
> 1. A landscape, the camera zooms in on an open field where a lone tree stands. A single tree, weathered by time, stands tall and proud, its branches reaching towds the heavens.
> 2. A landscape of the Middle East, a vast and desolate desert, with dunes of golden sand stretching to the horizon.
> 3. A landscape filled with vibrant colors and a sense of peace and hmony.

> **Input:** a woman
> **Generated prompts:**
> 1. a woman in the style of a classic disney animation, dk white and dk beige and yellow, lofi, lush scenery, iconic album covers, 8k resolution, detailed chacter design.
> 2. a woman in a white dress, in the background a lge window, from outside the window you can see the heavy rain and thunder, night, cinematic, foggy, mysterious, 8k, HD, Fujifi.
> 3. a woman 25 yes old, weing black t shirt and blue jeans with black sneakers walking in a train station.

> **Input:** Spiderman
> **Generated prompts:**
> 1. Spiderman, 4k, cinematic light, high production value, intricate details, high resolution.
> 2. Spiderman and his incredible strength and speed e on full display. The cityscape ound him is a blur of colors, from bright blue to purple.
> 3. Spiderman and his face is a canvas of shifting geometric patterns, his eyes spkling in the reflection of a neby window.

> **Input:** A Chinese dragon
> **Generated prompts:**
> 1. A Chinese dragon flies high in the sky, with the word ;dragon; printed on its body.
> 2. A Chinese dragon flies in a dk and mysterious universe, with a Chinese aesthetic.. a girl walking on the street. a 1970;s thouse film about a woman standing on a rock..
> 3. A Chinese dragon is flying in the air, and the camera lens is rotating ound the dragon.

## 7 Conclusion

This paper provides the first systematic research on text-to-video prompts. Specifically, we introduce VidProM, the first dataset comprising 1.67 million unique text-to-video prompts, 6.69 million videos generated by four state-of-the-art diffusion models, along with NSFW probabilities, 3072-dimensional prompt embeddings, and additional related metadata. To highlight the necessity of VidProM, we compare the basic information and the semantics of prompts of our VidProM to DiffusionDB, a text-to-image prompt dataset. Finally, we outline the potential research directions inspired by our VidProM, such as text-to-video prompt engineering, fake video detection, and video copy detection for diffusion models. We hope the curated large and diverse prompt-video dataset will advance research in the text-video domain.

**Limitation**. We recognize that the videos currently generated are short and not of the highest quality. In the future, we intend to enhance our dataset by incorporating high-quality videos produced by more advanced models, like Sora, using our long and detailed prompts. At the time of camera-ready version, we utilize three new powerful text-to-video models models to generate 10,000 videos with each for example: 10,000 videos of 8 seconds at 720p quality for StreamingT2V [58], 10,000 videos of 8 seconds at 720p quality for Open-Sora 1.2 [59], and 10,000 videos of 6 seconds at 720p quality for CogVideoX-2B [60]. They are publicly available at https://huggingface.co/datasets/WenhaoWang/VidProM/tree/main/example.

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

# A  Semantic De-duplication Algorithm

This section shows how we select semantically unique prompts, *i.e.*, for any two arbitrary prompts, their cosine similarity is less than $\theta = 0.8$. We first show our algorithm:

---

**Algorithm 1** Selecting Semantically Unique Prompts

---

**Input:** unique prompts $P_{unique} = \{p_1, p_2, ..., p_N\}$, prompt embeddings $\mathbf{E} \in \mathbb{R}^{N \times d}$, filtering threshold $\theta$

1: Compute similarity matrix $\mathbf{S} = \mathbf{E} \times \mathbf{E}^T$ and initialize $P_{similar} = \{\}$
2: **for** $i = 1$ to $N$ **do**
3:     **for** $j = i + 1$ to $N$ **do**
4:         **if** $S_{i,j} > \theta$ **then**
5:             Add $\{p_i, p_j\}$ to $P_{similar}$
6:         **end if**
7:     **end for**
8: **end for**
9: Initialize $P_{all} = \{\}$
10: **for** each pair $\{p_i, p_j\}$ in $P_{similar}$ **do**
11:     **if** $p_i$ not in $P_{all}$ **then**
12:         Add $p_i$ to $P_{all}$
13:     **end if**
14:     **if** $p_j$ not in $P_{all}$ **then**
15:         Add $p_j$ to $P_{all}$
16:     **end if**
17: **end for**
18: Initialize $P_{del} = \{\}$
19: **for** each pair $\{p_i, p_j\}$ in $P_{similar}$ **do**
20:     **if** $p_i$ and $p_j$ are both in $P_{all}$ **then**
21:         Add $p_i$ to $P_{del}$
22:     **end if**
23: **end for**
24: Initialize $P_{unique\_sem} = \{\}$
25: **for** each $p_i$ in $P_{unique}$ **do**
26:     **if** $p_i$ is not in $P_{del}$ **then**
27:         Add $p_i$ to $P_{unique\_sem}$
28:     **end if**
29: **end for**
**Output:** $P_{unique\_sem}$

---

Then we analyze the **efficiency** of the algorithm. The most time-consuming part of our algorithm lies in building $P_{similar}$, which includes calculating the similarity matrix and selecting pairs with a similarity greater than $\theta$. To complete this process with $N = 1,672,243$ and $d = 3072$, we distribute the workload across 8 A100 GPUs and 128 CPU cores using Faiss [61]. It only takes approximately 0.604 hours.

# B  Domain Gap between Video Captions and Text-to-Video Prompts

This section highlights the domain gap between training video captions and real user prompts. The domain gap may hinder the model's ability to generate satisfactory videos due to a lack of exposure to topics of interest to users and the style difference between captions and prompts. In Fig. 6, we randomly select 1.6 million captions from Panda-70M [18] and visualize them with our VidProM. We find that **their distributions are significantly different.** On one hand, **the covered topics are different.** For instance, users may want to generate videos of city nights or superheroes like Spider-Man. However, in the training data, there are few or no related videos on these topics. One possible solution is to filter out training videos that cover topics of human interest before training text-to-video models. On the other hand, **there may be a difference in styles of describing videos.** For instance, users may start a prompt with "Generate a video of" and end it with "4K". However,

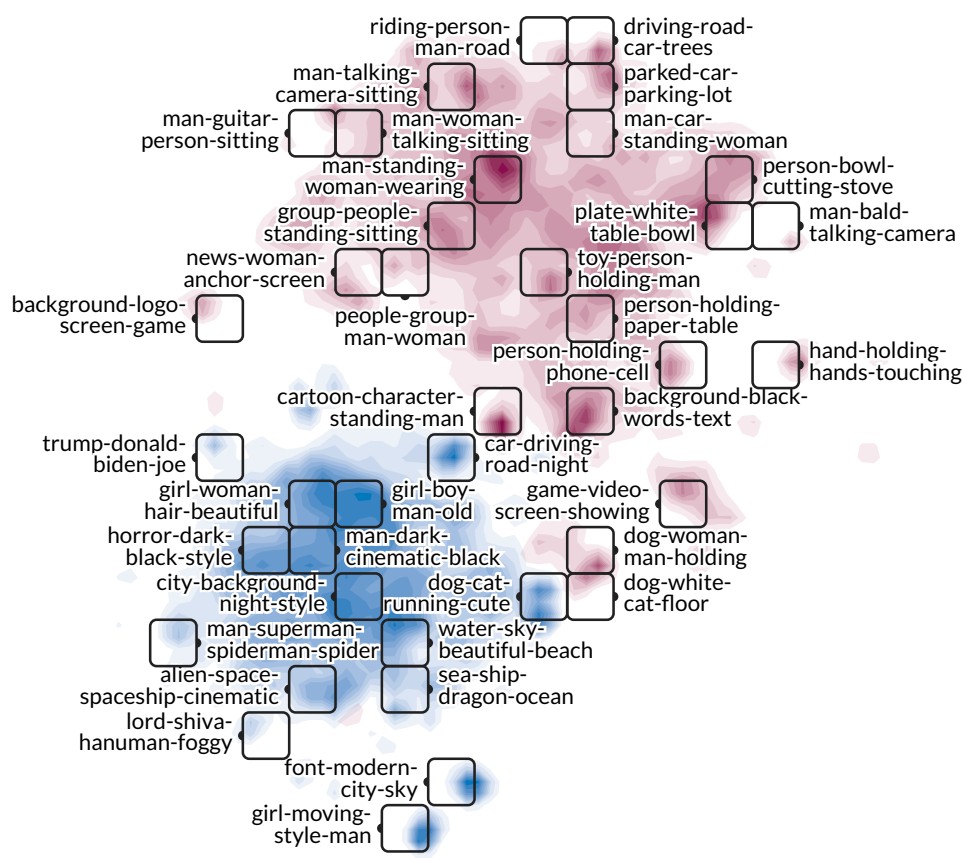

Figure 6: The WizMap [26] of our VidProM and Panda-70M [18]. Click here for an interactive one.

these words may not appear in the training captions. One suggestion is to train a text style transfer model [62, 63] to convert the style of the captions to that of the user prompts or reverse.

## C Benchmarking Existing Fake Image Detection on VidProM

This section shows the current fake image detection methods **fail** to generalize to fake video detection.

**Implementation details.** To benchmark existing fake image detection algorithms on our dataset, we test eight open-source, state-of-the-art models [42, 44, 43, 64, 46, 45, 47, 49] available at `https://github.com/Ekko-zn/AIGCDetectBenchmark`. We randomly select 10,000 videos generated by Pika [2], VideoCraft2 [4], Text2Video-Zero [3], and ModelScope [5], respectively, to serve as fake samples. Similarly, we choose 10,000 real videos randomly from DVSC2023 [65]. Since none of the state-of-the-art models can process entire videos directly, we extracted the middle frame from each video to use as the input image for each model. We evaluate the models using two metrics: Accuracy and Mean Average Precision (mAP).

**Experimental results.** The performance of the models is detailed in Tables 2 and 3. Our observations reveal that **all existing models struggle to differentiate between fake and real videos**: (1) Methods initially designed for detecting GAN-generated images, such as [42] and [46], are ineffective with videos produced by diffusion models, likely because they are trained on images from significantly different generative methods. (2) Techniques specifically developed for the diffusion process [47] or for generalizing across various generative models [49] also fail in detecting fake videos, indicating that the key features identified for spotting diffusion-generated images may not be as widely applicable as previously assumed. (3) Interestingly, methods that rely on traditional cues, such as global texture statistics [64] and frequency analysis [44], are the most effective. This underscores the continuing importance of traditional image processing knowledge.

Table 2: The accuracy of fake image detection methods on fake video detection task.

| Accuracy (%) | Pika | VideoCraft2 | Text2Video-Zero | ModelScope | **Average** |
|---|---|---|---|---|---|
| CNNSpot [42] | 51.17 | 50.18 | 49.97 | 50.31 | 50.41 |
| FreDect [44] | 50.07 | 54.03 | 69.88 | 69.94 | 60.98 |
| Fusing [43] | 50.60 | 50.07 | 49.81 | 51.28 | 50.44 |
| Gram-Net [64] | 84.19 | 67.42 | 52.48 | 50.46 | 63.64 |
| LGrad [46] | 53.73 | 51.75 | 41.05 | 60.22 | 51.69 |
| LNP [45] | 43.48 | 45.10 | 47.50 | 45.21 | 45.32 |
| DIRE [47] | 50.53 | 49.95 | 48.96 | 48.32 | 49.44 |
| UnivFD [49] | 49.41 | 48.65 | 49.58 | 57.43 | 51.27 |

Table 3: The mAP of fake image detection methods on fake video detection task.

| mAP (%) | Pika | VideoCraft2 | Text2Video-Zero | ModelScope | **Average** |
|---|---|---|---|---|---|
| CNNSpot [42] | 54.63 | 41.12 | 44.56 | 46.95 | 46.82 |
| FreDect [44] | 47.82 | 56.67 | 75.31 | 64.15 | 60.99 |
| Fusing [43] | 57.64 | 41.64 | 40.51 | 56.09 | 48.97 |
| Gram-Net [64] | 94.32 | 80.72 | 57.73 | 43.54 | 69.08 |
| LGrad [46] | 54.49 | 53.21 | 36.69 | 66.53 | 52.73 |
| LNP [45] | 44.28 | 44.08 | 46.81 | 39.62 | 43.70 |
| DIRE [47] | 49.21 | 50.44 | 44.52 | 48.64 | 48.20 |
| UnivFD [49] | 48.63 | 42.36 | 48.46 | 70.75 | 52.55 |

# D  Video Copy Detection for Diffusion Models

This sections shows whether the videos generated by text-to-video diffusion models replicate content of their training or existing videos.

## D.1  Experimental Setup

**The original videos for matching.** We randomly select 1 million videos from the training source of VideoCraft2 [4] and ModelScope [5] to analyze the extent to which generated videos replicate their training data. We acknowledge that using this source may result in an underestimation of the replication ratio, but it is sufficient for analyzing the phenomenon of replication.

**The model for video copy detection.** We select our previous winning solution, FCPL [53], as the model for video copy detection. It achieves state-of-the-art on DVSC2023 [65] while maintaining high efficiency.

## D.2  Observations

**Qualitative observations.** In Fig. 7, we demonstrate that text-to-video diffusion models can replicate the content from their training data or existing videos. This replication is acceptable for fair use purposes such as education, news reporting, and parody. However, misusing videos that contain replicated content with copyright could constitute an infringement. For example, in Fig. 7, the video generated by the commercial model Pika closely replicate the content of the famous painting, "The Persistence of Memory", which is copyrighted by the Gala-Salvador Dalí Foundation. If someone uses this generated video for profit without permission, it could potentially constitute copyright infringement.

**Quantitative observations.**

We calculate the maximum normalized score for each generated video by comparing it to all reference videos. As shown in Fig. 8, we count the number of generated videos with a maximum normalized score exceeding 0. Generally, a normalized score greater than 0 suggests a high likelihood of replicated content [66, 67, 65]. Our observations include: (1) Although replication occurs, it represents only a very small proportion of the generated videos. For instance, only about 2% of the videos generated by Text2Video-Zero have a maximum normalized score greater than 0. (2) The commercial model Pika has the lowest replication rate, while the open-source model Text2Video-Zero has the highest.

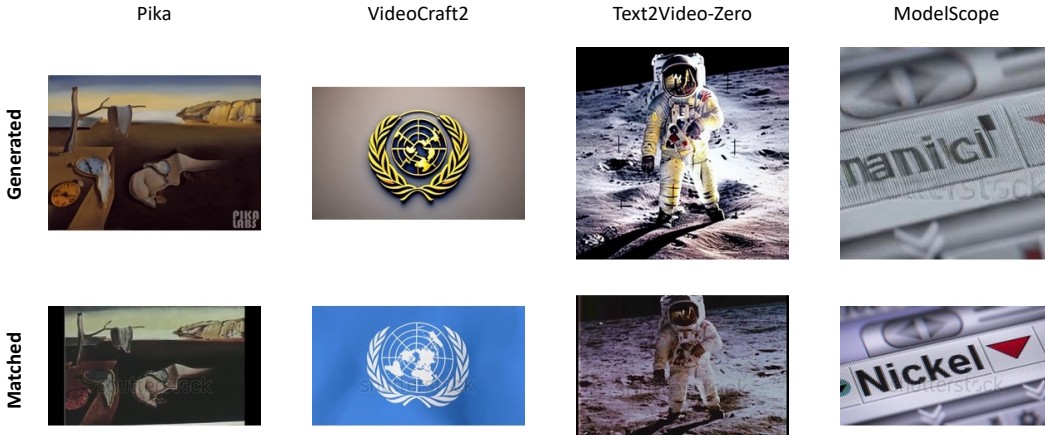

Figure 7: Some generated videos (top) by text-to-video diffusion models may replicate content of their training or existing videos (bottom). We use the matched frame in one video to represent the whole video.

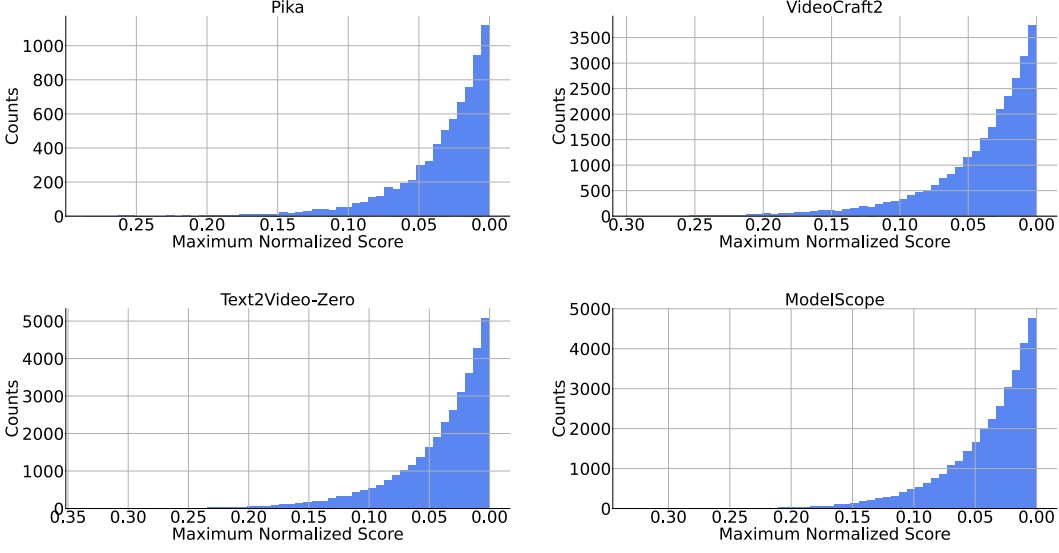

Figure 8: The counts of generated videos with different maximum normalized scores.

This difference may be partly because commercial models have implemented measures to prevent replication and mitigate related legal risks.

### D.3 Limitations and Future Directions

In Fig. 9, we visualize some failure cases gained by the current video copy detection models. We observe that, although the generated videos replicate content from existing ones, the video copy detection model fails to produce a normalized score higher than 0. The underlying issue is that these models are trained only on manually-generated transformations (as shown in Fig. 10), and therefore, they do not generalize well to replication patterns produced by diffusion models.

To establish video copy detection for diffusion models, a feasible approach is to utilize existing video copy detection systems as filters to identify potential pairs containing replicated content. Following this, human labelers can be employed to verify these pairs. The validated pairs can then be used to train a new detection model.

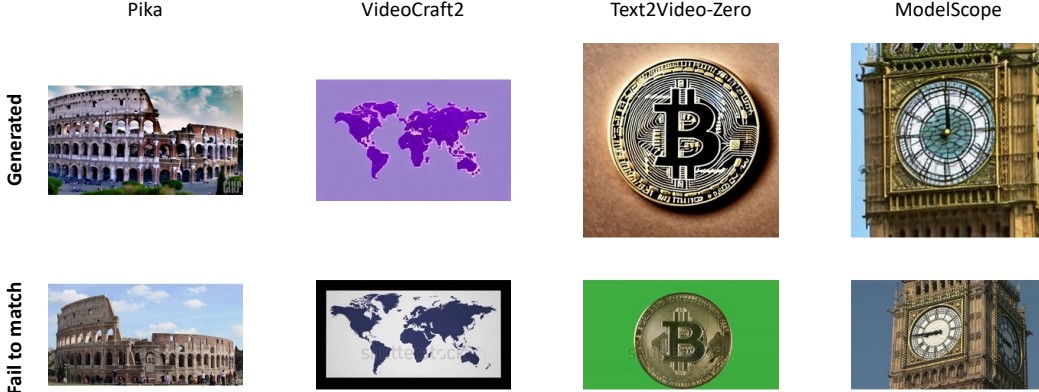

Figure 9: Using existing video copy detection models fails to regard these cases as replicated content.

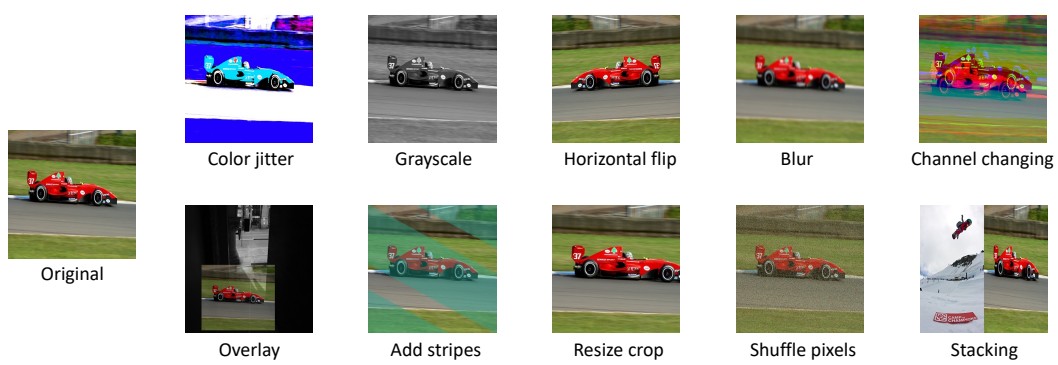

Figure 10: The transformations used to train current video copy detection models [51, 52, 53, 54].

# E    Data Sheet for VidProM

**For what purpose was the dataset created?** Was there a specific task in mind? Was there a specific gap that needed to be filled? Please provide a description.

The arrival of Sora marks a new era for text-to-video diffusion models, bringing significant advancements in video generation and potential applications. However, Sora, along with other text-to-video diffusion models, is highly reliant on prompts, and there is no publicly available dataset that features a study of text-to-video prompts. We underscore the need for a new prompt dataset specifically designed for text-to-video generation by illustrating how VidProM differs from DiffusionDB, a large-scale prompt-gallery dataset for image generation.

**Who created the dataset (e.g., which team, research group) and on behalf of which entity (e.g., company, institution, organization)?**

The dataset was created by Wenhao Wang (University of Technology Sydney) and Yi Yang (Zhejiang University).

**Who funded the creation of the dataset?** If there is an associated grant, please provide the name of the grantor and the grant name and number.

Funded in part by Faculty of Engineering and Information Technology Scholarship, University of Technology Sydney.

**Any other comments?**

None.

## Composition

**What do the instances that comprise the dataset represent (e.g., documents, photos, people, countries)?** Are there multiple types of instances (e.g., movies, users, and ratings; people and interactions between them; nodes and edges)? Please provide a description.

Each instance consists of a text-to-video prompt, a UUID, a timestamp, six NSFW probabilities, a 3072-dimensional prompt embedding, and four generated videos from Pika, VideoCraft2, Text2Video-Zero, and ModelScope.

**How many instances are there in total (of each type, if appropriate)?**

There are 1,672,243 instances in total.

**Does the dataset contain all possible instances or is it a sample (not necessarily random) of instances from a larger set?** If the dataset is a sample, then what is the larger set? Is the sample representative of the larger set (e.g., geographic coverage)? If so, please describe how this representativeness was validated/verified. If it is not representative of the larger set, please describe why not (e.g., to cover a more diverse range of instances, because instances were withheld or unavailable).

The dataset contains all possible instances up to February 2024.

**What data does each instance consist of?** "Raw" data (e.g., unprocessed text or images) or features? In either case, please provide a description.

Each instance consists of a text-to-video prompt, a UUID, a timestamp, six NSFW probabilities, a 3072-dimensional prompt embedding, and four generated videos from Pika, VideoCraft2, Text2Video-Zero, and ModelScope.

**Is there a label or target associated with each instance?** If so, please provide a description.

The labels associated with each video are the prompts and other related data.

**Is any information missing from individual instances?** If so, please provide a description, explaining why this information is missing (e.g., because it was unavailable). This does not include intentionally removed information, but might include, e.g., redacted text.

Everything is included. No data is missing.

**Are relationships between individual instances made explicit (e.g., users' movie ratings, social network links)?** If so, please describe how these relationships are made explicit.

Not applicable.

**Are there recommended data splits (e.g., training, development/validation, testing)?** If so, please provide a description of these splits, explaining the rationale behind them.

No. This dataset is not for ML model benchmarking. Researchers can use any subsets of it.

**Are there any errors, sources of noise, or redundancies in the dataset?** If so, please provide a description.

No. All videos and prompts are extracted as is from the Discord chat log.

**Is the dataset self-contained, or does it link to or otherwise rely on external resources (e.g., websites, tweets, other datasets)?**

The dataset is entirely self-contained.

**Does the dataset contain data that might be considered confidential (e.g., data that is protected by legal privilege or by doctor–patient confidentiality, data that includes the content of individuals' nonpublic communications)?** If so, please provide a description. Unknown to the authors of the datasheet.

It is possible that some prompts contain sensitive information. However, it would be rare, as the Pika Discord has rules against writing personal information in the prompts, and there are moderators removing messages that violate the Discord rules.

**Does the dataset contain data that, if viewed directly, might be offensive, insulting, threatening, or might otherwise cause anxiety?** If so, please describe why.

We collect videos and their prompts from the Pika discord server. Even though the discord server has rules against users sharing any NSFW (not suitable for work, such as sexual and violent content) and illegal images, VidProM still contains some NSFW videos and prompts that were not removed by the server moderators. To mitigate this, we provide six separate NSFW probabilities, allowing researchers to determine a suitable threshold for filtering out potentially unsafe data specific to their tasks.

**Does the dataset identify any subpopulations (e.g., by age, gender)?** If so, please describe how these subpopulations are identified and provide a description of their respective distributions within the dataset.

No.

**Is it possible to identify individuals (i.e., one or more natural persons), either directly or indirectly (i.e., in combination with other data) from the dataset?** If so, please describe how.

No.

**Any other comments?**

None.

---

| Collection |
|:---:|

**How was the data associated with each instance acquired?** Was the data directly observable (e.g., raw text, movie ratings), reported by subjects (e.g., survey responses), or indirectly inferred/derived from other data (e.g., part-of-speech tags, model-based guesses for age or language)? If the data was reported by subjects or indirectly inferred/derived from other data, was the data validated/verified? If so, please describe how.

The data was directly observed from the Pika Discord Channel. It was gathered from channels where users can generate videos by interacting with a bot, which consisted of messages of user generated videos and the prompts used to generate those images.

**What mechanisms or procedures were used to collect the data (e.g., hardware apparatuses or sensors, manual human curation, software programs, software APIs)?** How were these mechanisms or procedures validated?

The data was gathered using a DiscordChatExporter [23], which collected videos and chat messages from each channel specified. We then extracted and linked prompts to videos. Random videos and prompts were selected and manually verified to validate the prompt-video mapping.

**If the dataset is a sample from a larger set, what was the sampling strategy (e.g., deterministic, probabilistic with specific sampling probabilities)?**

VidProM does not sample from a larger set.

**Who was involved in the data collection process (e.g., students, crowdworkers, contractors) and how were they compensated (e.g., how much were crowdworkers paid)?**

No crowdworkers are needed in the data collection process.

**Over what timeframe was the data collected? Does this timeframe match the creation timeframe of the data associated with the instances (e.g., recent crawl of old news articles)?** If not, please describe the timeframe in which the data associated with the instances was created.

All messages were generated between July 2023 and February 2024. VidProM includes the generation timestamps of all videos.

**Were any ethical review processes conducted (e.g., by an institutional review board)?** If so, please provide a description of these review processes, including the outcomes, as well as a link or other access point to any supporting documentation.

There were no ethical review processes conducted.

**Did you collect the data from the individuals in question directly, or obtain it via third parties or other sources (e.g., websites)?**
The data was directly obtained from individual messages in the Discord server.

**Were the individuals in question notified about the data collection?** If so, please describe (or show with screenshots or other information) how notice was provided, and provide a link or other access point to, or otherwise reproduce, the exact language of the notification itself.
Users of the channel were not notified about this specific gathering of data but agree to open-source their input and output under the Creative Commons Noncommercial 4.0 Attribution International License (CC BY-NC 4.0). The exact language is as follows:

> You hereby grant Mellis and other users a license to any of your Inputs and Outputs that you make available to other users on the Service under the Creative Commons Noncommercial 4.0 Attribution International License (as accessible here: https://creativecommons.org/licenses/by-nc/4.0/legalcode).

**Did the individuals in question consent to the collection and use of their data?** If so, please describe (or show with screenshots or other information) how consent was requested and provided, and provide a link or other access point to, or otherwise reproduce, the exact language to which the individuals consented.
By using the server and tools, users consented to the regulations posed by Pika Lab, the company that both made Pika text-to-video model and runs the Discord server. This implies consent by using the tool. The exact wording is as follows:

> You hereby grant Mellis and other users a license to any of your Inputs and Outputs that you make available to other users on the Service under the Creative Commons Noncommercial 4.0 Attribution International License (as accessible here: https://creativecommons.org/licenses/by-nc/4.0/legalcode).

**If consent was obtained, were the consenting individuals provided with a mechanism to revoke their consent in the future or for certain uses?** If so, please provide a description, as well as a link or other access point to the mechanism (if appropriate).
Users will have the option to report harmful content or withdraw videos they created by emailing Wenhao Wang (wangwenhao0716@gmail.com).

**Has an analysis of the potential impact of the dataset and its use on data subjects (e.g., a data protection impact analysis) been conducted?** If so, please provide a description of this analysis, including the outcomes, as well as a link or other access point to any supporting documentation.
No analysis has been conducted.

**Any other comments?**
None.

---

## Preprocessing

**Was any preprocessing/cleaning/labeling of the data done (e.g., discretization or bucketing, tokenization, part-of-speech tagging, SIFT feature extraction, removal of instances, processing of missing values)?** If so, please provide a description. If not, you may skip the remaining questions in this section.
No, we provide the original prompts and generated videos.

**Was the "raw" data saved in addition to the preprocessed/cleaned/labeled data (e.g., to support unanticipated future uses)?** If so, please provide a link or other access point to the "raw" data.
Yes, raw data is saved.

**Is the software that was used to preprocess/clean/label the data available?** If so, please provide a link or other access point.

Not applicable.

**Any other comments?**
None.

---

# Uses

---

**Has the dataset been used for any tasks already?** If so, please provide a description.
No.

**Is there a repository that links to any or all papers or systems that use the dataset?** If so, please provide a link or other access point.
No.

**What (other) tasks could the dataset be used for?**
This dataset can be used for video generative model evaluation, text-to-video diffusion model development, text-to-video prompt engineering, efficient video generation, fake video detection, video copy detection for diffusion models, and multimodal learning from synthetic videos.

**Is there anything about the composition of the dataset or the way it was collected and preprocessed/cleaned/labeled that might impact future uses?** For example, is there anything that a dataset consumer might need to know to avoid uses that could result in unfair treatment of individuals or groups (e.g., stereotyping, quality of service issues) or other risks or harms (e.g., legal risks, financial harms)? If so, please provide a description. Is there anything a dataset consumer could do to mitigate these risks or harms?
There is minimal risk for harm: the data were already public. Personally identifiable data (e.g., discord usernames) were not collected.

**Are there tasks for which the dataset should not be used?** If so, please provide a description.
All tasks that utilize this dataset should follow the licensing policies posed by Pika Lab.

**Any other comments?**
None.

---

# Distribution

---

**Will the dataset be distributed to third parties outside of the entity (e.g., company, institution, organization) on behalf of which the dataset was created?** If so, please provide a description.
Yes, the dataset is publicly available on the internet.

**How will the dataset will be distributed (e.g., tarball on website, API, GitHub)?** Does the dataset have a digital object identifier (DOI)?
The dataset is distributed on the project website: https://vidprom.github.io/. The dataset shares the same DOI as this paper.

**When will the dataset be distributed?**
The dataset is released on March 12nd, 2024.

**Will the dataset be distributed under a copyright or other intellectual property (IP) license, and/or under applicable terms of use (ToU)?** If so, please describe this license and/or ToU, and provide a link or other access point to, or otherwise reproduce, any relevant licensing terms or ToU, as well as any fees associated with these restrictions.
The prompts and videos generated by Pika in our VidProM are licensed under the CC BY-NC 4.0 license from the Pika Discord, allowing for non-commercial use with attribution. Additionally, similar to their original repositories, the videos from VideoCraft2, Text2Video-Zero, and ModelScope

are released under the Apache license, the CreativeML Open RAIL-M license, and the CC BY-NC 4.0 license, respectively.

**Have any third parties imposed IP-based or other restrictions on the data associated with the instances?** If so, please describe these restrictions, and provide a link or other access point to, or otherwise reproduce, any relevant licensing terms, as well as any fees associated with these restrictions.
No.

**Do any export controls or other regulatory restrictions apply to the dataset or to individual instances?** If so, please describe these restrictions, and provide a link or other access point to, or otherwise reproduce, any supporting documentation.
No.

**Any other comments?**
None.

---

## Maintenance

**Who will be supporting/hosting/maintaining the dataset?**
The authors of this paper will be supporting and maintaining the dataset.

**How can the owner/curator/manager of the dataset be contacted (e.g., email address)?**
The contact information of the curators of the dataset is listed on the project website: https://vidprom.github.io/.

**Is there an erratum?** If so, please provide a link or other access point.
There is no erratum for our initial release. Errata will be documented in future releases on the dataset website.

**Will the dataset be updated (e.g., to correct labeling errors, add new instances, delete instances)?** If so, please describe how often, by whom, and how updates will be communicated to dataset consumers (e.g., mailing list, GitHub)?
Yes, we will monitor cases when users can report harmful videos and creators can remove their videos. We will update the dataset bimonthly. Updates will be posted on the project website https://vidprom.github.io/. In the future, we intend to enhance our dataset by incorporating high-quality videos produced by more advanced models, like Sora, using our detailed prompts.

**If the dataset relates to people, are there applicable limits on the retention of the data associated with the instances (e.g., were the individuals in question told that their data would be retained for a fixed period of time and then deleted)?** If so, please describe these limits and explain how they will be enforced.
People can write an email to remove specific instances from VidProM.

**Will older versions of the dataset continue to be supported/hosted/maintained?** If so, please describe how. If not, please describe how its obsolescence will be communicated to dataset consumers.
We will continue to support older versions of the dataset.

**If others want to extend/augment/build on/contribute to the dataset, is there a mechanism for them to do so?** If so, please provide a description. Will these contributions be validated/verified? If so, please describe how. If not, why not? Is there a process for communicating/distributing these contributions to dataset consumers? If so, please provide a description.
Anyone can extend/augment/build on/contribute to VidProM. Potential collaborators can contact the dataset authors.

**Any other comments?**
None.

