# OpenReview forum: "VidProM: A Million-scale Real Prompt-Gallery Dataset for Text-to-Video Diffusion Models"
_NeurIPS.cc/2024/Datasets_and_Benchmarks_Track — NeurIPS 2024 Track Datasets and Benchmarks Poster_

### Official Review · Reviewer_iKQy · 2024-06-30
**Clear Accept**

**Rating:** 9
**Confidence:** 5
**Correctness:** yes
**Clarity:** yes

**Review:**

as above

**Strengths:**

as above

**Additional Feedback:**

no

**Documentation:**

yes

**Limitations:**

no as far as I know

**Opportunities For Improvement:**

One suggestion is that the GitHub repository is almost empty. The authors should provide scalable dataloaders, including torch.dataloader and webdataset, to further encourage potential research.

**Relation To Prior Work:**

yes

**Summary And Contributions:**

This paper is timely given the longstanding interest in video generation. It demonstrates the use of curation, statistics, and comparisons with other image-based benchmarks. Ultimately, the authors propose several potential research directions. The website is well-presented, leaving no further questions.

---

> ### Comment · Reviewer_iKQy · 2024-08-14
>
> Sorry for the short review. I couldn't find significant issues, but I will engage more in the later discussion.

---

> > ### Author Response · Authors · 2024-08-14
> >
> > Thank you very much for your review. We are currently finalizing the rebuttal process for the other reviewers, and following your suggestions, we have written the suggested code. We will upload the responses to the other reviewers by August 16, 2024, AOE.

---

> ### Author Rebuttal · Authors · 2024-08-14
>
> *We sincerely thank you for your highly positive feedback and helpful suggestions. We address your questions below.*
>
> **Q1. One suggestion is that the GitHub repository is almost empty. The authors should provide scalable dataloaders, including torch.dataloader and webdataset, to further encourage potential research.**
>
> A1. We thank you for this constructive suggestion. According to this advice, we provide the code for using `torch.dataloader` and `webdataset` to load our VidProM in our GitHub repository: https://github.com/WangWenhao0716/VidProM.
>
> We also list the code below for a clearer demonstration.
>
> ----------------------------
>
> # Dataloader
> We use the `example` folder to illustrate how to load VidProM using [PyTorch Dataloader](https://pytorch.org/tutorials/beginner/basics/data_tutorial.html) and [WebDataset](https://github.com/webdataset/webdataset).
>
> ## PyTorch Dataloader
>
> The `example` directory is
> ```
> *example
>     *VidProM_unique_example.csv
>     *VidProM_embed_example.hdf5
>     *pika_videos_example
> 	pika-xxx-xxx.mp4
>             pika-xxx-xxx.mp4
> 	...
>     *t2vz_videos_example
> 	t2vz-xxx-xxx.mp4
>             t2vz-xxx-xxx.mp4
> 	...
>     *vc2_videos_example
> 	vc2-xxx-xxx.mp4
>             vc2-xxx-xxx.mp4
> 	...
>     *ms_videos_example
> 	ms-xxx-xxx.mp4
>             ms-xxx-xxx.mp4
> 	...
> ```
>
> We have the following PyTorch Dataloader:
> ```python
> import os
> import pandas as pd
> import h5py
> import torch
> from torch.utils.data import Dataset, DataLoader
> from torchvision.io import read_video
> import numpy as np
> ```
>
>
> ```python
> class VidProMDataset(Dataset):
>     def __init__(self, csv_file, hdf5_file, video_dirs, transform=None):
>
>         self.metadata = pd.read_csv(csv_file)
>         self.video_dirs = video_dirs
>         self.transform = transform
>         self.nsfw_names = ['toxicity','obscene','identity_attack','insult','threat','sexual_explicit']
>
>         self.hdf5_file =  h5py.File(hdf5_file, 'r')
>         self.hdf5_uuid = np.array(self.hdf5_file["uuid"][:], dtype=object).astype(str).tolist()
>         self.hdf5_embed = np.array(self.hdf5_file['embeddings'])
>
>     def __len__(self):
>         return len(self.metadata)
>
>     def __getitem__(self, idx):
>
>         video_info = self.metadata.iloc[idx]
>         video_id = video_info['uuid']
>         prompt = video_info['prompt']
>         time = video_info['time']
>         nsfw_scores = torch.tensor(list(video_info[self.nsfw_names]))
>
>         embed = torch.tensor(self.hdf5_embed[self.hdf5_uuid.index(video_id)])
>         video_path = self._find_video_path(video_id)
>         video_frames, _, _ = read_video(video_path, pts_unit='sec')
>
>         if self.transform:
>             video_frames = self.transform(video_frames)
>
>         return {
>             'video_id': video_id,
>             'video_frames': video_frames,
>             'embed': embed,
>             'prompt': prompt,
>             'time': time,
>             'nsfw_scores': nsfw_scores
>         }
>
>     def _find_video_path(self, video_id):
>         for video_dir in self.video_dirs:
>             video_file = os.path.join(video_dir, video_dir.split('_')[0] + f"-{video_id}.mp4")
>             if os.path.exists(video_file):
>                 return video_file
>         raise FileNotFoundError(f"Video {video_id}.mp4 not found in any of the directories.")
>
>     def __del__(self):
>         self.hdf5_file.close()
>
> ```
>
> ```python
> csv_file = 'VidProM_unique_example.csv'
> hdf5_file = 'VidProM_embed_example.hdf5'
> video_dirs = ['t2vz_videos_example', 'pika_videos_example', 'vc2_videos_example', 'ms_videos_example']
> dataset = VidProMDataset(csv_file, hdf5_file, video_dirs)
> dataloader = DataLoader(dataset, batch_size=16, shuffle=False, num_workers=0)
> ```
>
> ## WebDataset
>
> We can load videos using WebDataset from the `tar` files directly, and we assume the directory is
> ```
> *example
>     *VidProM_unique_example.csv
>     *VidProM_embed_example.hdf5
>     *pika_videos_example.tar
>     *t2vz_videos_example.tar
>     *vc2_videos_example.tar
>     *ms_videos_example.tar
> ```
> We have the following:
>
> ```python
> import os
> import io
> import av
> import pandas as pd
> import h5py
> import numpy as np
> from PIL import Image
> import torchvision.transforms as transforms
> import torch
> import webdataset as wds
> ```
>
>
> ```python
> tar_file_path = 't2vz_videos_example.tar' # we use t2vz_videos_example.tar for example
> csv_file = 'VidProM_unique_example.csv'
> hdf5_file = 'VidProM_embed_example.hdf5'
> dataset = wds.WebDataset(tar_file_path)
> metadata = pd.read_csv(csv_file)
> hdf5_file = h5py.File(hdf5_file, 'r')
> hdf5_uuid = np.array(hdf5_file["uuid"][:], dtype=object).astype(str).tolist()
> hdf5_embed = np.array(hdf5_file['embeddings'])
> ```
>
> ```python
> for sample in dataset:
>     #obtain tensor of a video
>     binary_data = sample['mp4']
>     container = av.open(io.BytesIO(binary_data))
>     transform = transforms.ToTensor()
>     frames = []
>     for frame in container.decode(video=0):
>         img = frame.to_image()
>         img_tensor = transform(img)
>         frames.append(img_tensor)
>     video_tensor = torch.stack(frames)
>
>     #obtain uuid of a video
>     uuid = '-'.join(sample['__key__'].split('/')[-1].split('-')[1:])
>
>     #obtain the prompt
>     prompt = list(metadata[metadata['uuid']==uuid].iloc[:, 1])[0]
>
>     #obtain the time
>     time = list(metadata[metadata['uuid']==uuid].iloc[:, 2])[0]
>
>     #obtain the nsfw_scores
>     nsfw_scores = list(metadata[metadata['uuid']==uuid].iloc[0, 3:])
>
>     #obtain the prompt embedding
>     embed = torch.tensor(hdf5_embed[hdf5_uuid.index(uuid)])
>
> ```

---

> > ### Comment · Reviewer_iKQy · 2024-08-14
> >
> > thank you

---

### Official Review · Reviewer_rjWb · 2024-07-23
**Review of "VidProM: A Million-scale Real Prompt-Gallery Dataset for Text-to-Video Diffusion Models"**

**Rating:** 6
**Confidence:** 3
**Correctness:** Yes.
**Clarity:** Yes.

**Review:**

The paper is easy to understand and effectively curates a dataset. It clearly explains four aspects of the process: 1) Collecting Source HTML Files, 2) Extracting and Embedding Prompts, 3) Assigning NSFW Probabilities, and 4) Scraping and Generating Videos. The proposed pipeline for creating the dataset is well-structured and justifies the need for VidProM.

However, the paper would benefit from a comparison between the distribution of synthetic data and real-world data. An ablation study should also be conducted to evaluate the usage of the generated data. Simply checking the range of data distribution is insufficient. Additionally, considering that PIKA has many generated videos that are unrealistic, it would be beneficial for the authors to demonstrate that VidProM provides improvements over previous approaches.

It would also be interesting to investigate whether longer prompts yield better results than shorter prompts, as this could provide valuable insights into the effectiveness of prompt length in text-to-video generation.

**Strengths:**

The paper is well-written, and the motivation is clearly articulated.

**Additional Feedback:**

N/A.

**Documentation:**

Yes.

**Ethics:**

N/A.

**Limitations:**

Yes.

**Opportunities For Improvement:**

Synthetic data's reliability is not yet guaranteed. More ablation studies should be conducted to verify its efficiency. Additionally, it is important to evaluate the bias between real-world and synthetic videos.

**Relation To Prior Work:**

Yes.

**Summary And Contributions:**

This paper proposes a new prompt dataset, VidProM, specifically designed for text-to-video generation. VidProM stands out from DiffusionDB as it is the first dataset of its kind, serving as a text-to-video prompt gallery. The authors also offer insights into several new research directions:

1) Improving Models: Researchers can use VidProM as a comprehensive set of prompts to evaluate their trained models, refine new models using prompt-generated video pairs, and explore prompt engineering. 2) Enhancing Efficiency: Researchers can search for related prompts within VidProM and reconstruct new videos from similar existing ones, thus avoiding the need to generate videos from scratch. 3) Ensuring Safety: Researchers can develop specialized models to differentiate generated videos from real ones to combat misinformation and create video copy detection models to address potential copyright issues.

---

> ### Author Rebuttal · Authors · 2024-08-15
>
> *We sincerely appreciate your positive feedback and helpful suggestions. We hope that after addressing the following concerns, our paper will further meet your expectations and potentially receive a higher score.*
>
> **Q1. The bias between real-world and synthetic videos. The improvements brought by VidProM over previous approaches.**
>
> A1. Thank you for the question, and we apologize for this misunderstanding. As shown in Fig. 1 of the attached PDF, the distributions of *synthetic data* and *real-world data* may *differ to some degree*. However, ***`currently,`*** our paper primarily focuses on ***inspiring new research*** related to ***text-to-video generation***, rather than on **designing or using synthetic data** to **replace real data or improve the performance of existing methods, such as multimodal learning**. ***Most of the inspired new tasks are not or only slightly related to the reliability/bias/realism of synthetic data compared to real-world data.*** Specifically, we outline the roadmap below, in ***`chronological`*** order, to demonstrate how our VidProM ***inspires new research***.
>
> ### ***`Currently Feasible Tasks`***
>
> The currently feasible tasks include **Video Generative Model Evaluation**, **Text-to-Video Prompt Engineering**, **Fake Video Detection**, and **Video Copy Detection for Diffusion Models**.
>
> **Video Generative Model Evaluation:** **`Based on our VidProM dataset`**, recent work like VIDEOSCORE [1] establishes automated metrics to simulate fine-grained human feedback for video generation. Additionally, Liao Mingxiang et al. [2] recently evaluated text-to-video generation models from a dynamics perspective. **These works underscore the importance of our VidProM.**
>
> **Text-to-Video Prompt Engineering:** As shown in the Supplementary (Section C), we train a large language model on VidProM for **automatic text-to-video prompt completion**. People can use this model and any text-to-video diffusion models to generate video content by providing **only a few words**.
>
> **Fake Video Detection:** **`Based on our VidProM dataset`**, recently, Ji Lichuan et al. [3] introduced an extensive video dataset designed specifically for AI-Generated Video Detection.
>
> **Video Copy Detection for Diffusion Models:**  As detailed in Supplementary (Section E), we discuss the copyright issues associated with text-to-video diffusion models, the challenges faced by current video copy detection models, and how the proposed VidProM can be used to address these challenges.
>
> ***The application of our VidProM for fake video detection and video copy detection in diffusion models highlights its importance in AI safety.***
>
> ### ***`Short-Term Achievable Tasks`***
>
> The short-term achievable tasks include **Text-to-Video Diffusion Model Development** and **Efficient Video Generation**. We acknowledge that videos generated by current text-to-video diffusion models are *short* and *not of the highest quality*. However, as these diffusion models advance, they will be capable of generating longer and higher-quality videos. This will enable researchers to either *distill new text-to-video diffusion models using prompt-generated video pairs* or *reconstruct videos directly from existing ones* ***with the prompts in VidProM***. Also see **Q1** of `Reviewer 74gs`.
>
> ### ***`Long-Term  Achievable Tasks`***
>
> The long-term achievable tasks include **Multimodal Learning from Synthetic Videos**. ***We notice that the reviewer’s concern about the difference/bias between synthetic and real-world data may come from this argument, and we apologize again here.*** In the paper, we argue that our VidProM “***offers a promising research direction*** for tasks like video-text retrieval and video captioning,” implying that this potential may be realized **in the future**, rather than **immediately**, due to the *biases* and *unrealism* issues raised by the reviewer. As text-to-video diffusion models advance to become more realistic, like the ***World Simulator*** that Sora aims to be, this goal could eventually be achieved. At that time, researchers may use the 1.67 million prompts and ***World Simulators*** to generate videos to substitute real-world data. ***We will revise our paper accordingly to prevent any further misunderstandings.***
>
> [1] Xuan, He, et al. "VIDEOSCORE: Building Automatic Metrics to Simulate Fine-grained Human Feedback for Video Generation." arXiv:2406.15252 (2024).
>
> [2] Liao, Mingxiang, et al. "Evaluation of Text-to-Video Generation Models: A Dynamics Perspective." arXiv:2407.01094 (2024).
>
> [3] Ji, Lichuan, et al. "Distinguish Any Fake Videos: Unleashing the Power of Large-scale Data and Motion Features." arXiv:2405.15343 (2024).
>
> **Q2. It would also be interesting to investigate whether longer prompts yield better results than shorter prompts.**
>
> A2. We thank this insightful question and provide the investigation we have done:
>
> - ***Compared to using short prompts, videos generated by long and detailed prompts are more specific, dynamic, and story-driven.*** We provide a few visual comparisons in the Fig. 2 of the attached PDF.
> - ***The developers of most advanced text-to-video models prefer long prompts.*** For example, ***most*** of the videos displayed on the official websites of [CogVideoX](https://github.com/THUDM/CogVideo) and [Sora](https://openai.com/index/sora/) are generated using long prompts. Specifically, the prompt for the famous ‘Tokyo Girl’ video contains 64 words, while the longest prompt on the OpenAI official website comprises 95 words.
> - ***The effectiveness of long prompts has been verified in a related domain, i.e. text-to-image.*** For instance, **ChatGPT** will automatically translate “*draw a cat*” to “*A realistic and detailed depiction of a domestic cat sitting in a sunlit living room. The cat is fluffy, with a mixture of gray and white fur, and bright green eyes. It is perched comfortably on a soft beige couch, surrounded by a few colorful cushions.*” for **DALL·E 3**.

---

> > ### Comment · Reviewer_rjWb · 2024-08-25
> > **Review of "VidProM: A Million-scale Real Prompt-Gallery Dataset for Text-to-Video Diffusion Models"**
> >
> > Thank you for your thoughtful response. While I remain cautious about this "potential direction", I will keep my score unchanged.

---

> > > ### Author Response · Authors · 2024-08-25
> > >
> > > Thanks again for your reply. According to your suggestion, we will largely weaken or delete this potential direction in the camera-ready version.

---

### Official Review · Reviewer_74gs · 2024-07-24
**A large-scale dataset for text-to-video diffusion models**

**Rating:** 9
**Confidence:** 5
**Correctness:** Yes.

**Review:**

- A key advantage of this dataset is the inclusion of 1.67 million text-to-video prompts from real users. This data can be used to propose the foundation model for text-to-video generation, which can lead to more accurate and contextually relevant video generation.
- The quality of VidProM's prompts is quite good, consisting of complete sentences that can convey meaning, unlike DiffusionDB's prompts, which consist of a list of related words.
- The 6.69 million videos in VidProM are generated by current diffusion models such as Pika, VideoCraft2, Text2Video-Zero and ModelScope. Therefore, the video generation models using VidProM dataset are highly dependent on the quality of the current diffusion models. This does not give us the certainty that we can generate higher quality videos with this dataset.
- The videos in VidProM are 3.0 seconds or shorter - this could be another limitation of this dataset. Therefore, the authros need to consider whether it is possible to generate video longer than 3.0 seconds when the model is trained with VidProM.

**Strengths:**

- The motivation for proposing such a large text-video dataset is well-founded.
- According to the main text, the data collection and associated research areas appear reasonable and technically sound.
- The extensive analysis of the dataset, especially the prompts in VidProM compared to DiffusionDM, is very interesting and clearly confirms the need for the proposal of VidProM dataset.
- The paper is well-organized and clearly written.

**Additional Feedback:**

None.

**Clarity:**

- The paper is well written.

**Documentation:**

- The repository instructions are clear.

**Ethics:**

- I have no suspicions of ethical concerns.

**Limitations:**

- The short duration and quality of the videos in the proposed dataset

**Opportunities For Improvement:**

- The duration of the videos in VidProM should be specified more precisely for users.
- The authors needs to describe in detail the compositional relationship of the videos produced by the four SOTA generation models. Providing this information would offer valuable insights into the structure of the dataset and possible biases.

**Relation To Prior Work:**

- In this study, a comprehensive comparison between DiffusionDB was conducted, which shows the novelty of the proposed VidProM.

**Summary And Contributions:**

This paper introduces VidProM, a new large-scale dataset for text-to-video generation. In particular, VidProM is the first dataset that contains 1.67 million text-to-video prompts from real users. Moreover, it contains 6.69 million videos generated by four state-of-the-art diffusion models. This dataset can be employed for various research areas, such as text-to-video generation, efficient video generation, and video copy detection for diffusion models.

---

> ### Author Rebuttal · Authors · 2024-08-15
>
> *We sincerely thank you for your highly positive feedback and helpful suggestions. We address your questions below.*
>
> **Q1. The short duration and quality of the videos in the proposed dataset. This does not give us the certainty that we can generate higher quality videos with this dataset. The authors need to consider whether it is possible to generate video longer than 3.0 seconds when the model is trained with VidProM.**
>
> A1. We appreciate your insightful question. The videos in our VidProM are indeed short and not of the highest quality because *there were ***no*** publicly available text-to-video diffusion models that could meet this requirement at the time VidProM was being developed (the deadline of NeurIPS).* However, this issue can be addressed by a simple extension of VidProM. Specifically, based on the ***high-quality prompts in VidProM*** and the ***increasingly powerful text-to-video models becoming available over time***, researchers are able to generate high-quality, long videos on their own. By training on these newly generated videos, researchers can generate higher quality and longer videos with VidProM.
>
> For instance, currently, there are three more powerful text-to-video models:
>
> - [StreamingT2V](https://github.com/Picsart-AI-Research/StreamingT2V) can generate videos up to **120** seconds in length with **720p** quality.
> - [Open-Sora 1.2](https://github.com/hpcaitech/Open-Sora) can generate videos up to **16** seconds in length with **720p** quality.
> -  [CogVideoX-2B](https://github.com/THUDM/CogVideo) can generate videos up to **6** seconds in length with **720p** quality.
>
> To balance inference speed, video length, and video quality, we utilize these three different models to generate 10,000 videos with each for example: 10,000 videos of 8 seconds at 720p quality, 10,000 videos of 8 seconds at 720p quality, and 10,000 videos of 6 seconds at 720p quality. We have uploaded the videos generated by StreamingT2V and CogVideoX-2B to https://huggingface.co/datasets/WenhaoWang/VidProM/tree/main/example. Due to limited computational resources and the time-consuming generation process, we will upload the videos generated by Open-Sora 1.2 before August 24, 2024.
>
>
> **Q2. The duration of the videos in VidProM should be specified more precisely for users.**
>
> Thanks for this suggestion. The video lengths are as follows: **3.0** seconds for Pika, **1.6** seconds for VideoCraft2, **2.0** seconds for Text2Video-Zero, and **2.0** seconds for ModelScope. We will highlight these in the camera-ready version.
>
> **Q3. The authors needs to describe in detail the compositional relationship of the videos produced by the four SOTA generation models. Providing this information would offer valuable insights into the structure of the dataset and possible biases.**
>
> Thank you for the suggestion, and we apologize for not being entirely sure about what ‘the compositional relationship’ refers to.
>
> If the reviewer refers to the basic compositional relationship: **Each diffusion model generates the same number of videos.** Specifically, our VidProM has **1,672,243** unique prompts, resulting in the generation of **1,672,243** videos using Pika, **1,672,243** videos using VideoCraft2, **1,672,243** videos using Text2Video-Zero, and **1,672,243** videos using ModelScope. We will highlight these points in the camera-ready version.
>
> We also notice that a recent wonderful paper [1] focuses on comprehensive compositional relationships, including *consistent attribute binding*, *dynamic attribute binding*, *spatial relationships*, *action binding*, *object interactions*, and *generative numeracy*. Its current version features seven categories with **700 text prompts generated by GPT-4**, and we believe that with our **million-scale, real-user-generated prompts**, there will be new research opportunities and valuable insights.
>
> [1] Sun, Kaiyue, et al. "T2V-CompBench: A Comprehensive Benchmark for Compositional Text-to-video Generation." arXiv:2407.14505 (2024).

---

> > ### Author Response · Authors · 2024-08-20
> >
> > Dear Reviewer,
> >
> > As promised in the rebuttal, we have uploaded the long (8s) and high-quality (720p) videos generated by the Open-Sora: https://huggingface.co/datasets/WenhaoWang/VidProM/blob/main/example/opensora_videos_example.tar
> >
> > Thanks again for reviewing our paper.
> >
> > Authors of Submission #5

---

### Author Rebuttal · Authors · 2024-08-15

**Thanks**

We sincerely thank the ACs and reviewers for their dedicated efforts in reviewing our paper. We also thank all reviewers for their positive, thoughtful, and helpful feedback, and we will add all these suggestions to the final version of our paper.

***First of all, we sincerely appreciate that `Reviewer 74gs` and `Reviewer iKQy` give us **9** rating with **5** confidence***. We are also encouraged that `Reviewer 74gs` found “*The quality of VidProM's prompts is quite good*”, “*the data collection and associated research areas appear reasonable and technically sound*”, and “*The extensive analysis of the dataset is very interesting and clearly confirms the need for the proposal of VidProM dataset.*” We are also glad that `Reviewer rjWb` recognized our work as “*The authors also offer insights into several new research directions*” and “*The paper is well-written, and the motivation is clearly articulated.*” We are also happy that `Reviewer iKQy`  regards our “*website is well-presented, leaving no further questions*” and is willing to “*engage in discussion*”.

We have thoroughly addressed all the concerns raised by the reviewers in the below separate responses. Please let us know if you have any additional questions or concerns. We are happy to provide clarification.

Authors of Submission #5

---

### Decision · Program_Chairs · 2024-09-26

**Decision:**

Accept (Poster)

**Comment:**

All three reviewers are positive about this paper, of which two reviewers gave a high score 9. The third reviewer has some questions. The authors' rebuttal gave reasonable responses, mostly answered the questions. I concur this is a good paper on a difficult problem and it's timely. Between oral and poster perhaps a highlight.